# Inhibition of Cytomegalovirus by *Pentacta pygmaea* Fucosylated Chondroitin Sulfate Depends on Its Molecular Weight

**DOI:** 10.3390/v15040859

**Published:** 2023-03-28

**Authors:** Poonam Sharma, Rohini Dwivedi, Priya Ray, Jayanti Shukla, Vitor H. Pomin, Ritesh Tandon

**Affiliations:** 1Center for Immunology and Microbial Research, Department of Cell and Molecular Biology, University of Mississippi Medical Center, 2500 North State Street, Jackson, MS 39216, USA; 2Department of Biomolecular Sciences, University of Mississippi, Oxford, MS 38655, USA; 3Department of Medicine, University of Mississippi Medical Center, 2500 North State Street, Jackson, MS 39216, USA

**Keywords:** herpesviruses, marine sulfated glycans, virus entry

## Abstract

Many viruses attach to host cells by first interacting with cell surface proteoglycans containing heparan sulfate (HS) glycosaminoglycan chains and then by engaging with specific receptor, resulting in virus entry. In this project, HS–virus interactions were targeted by a new fucosylated chondroitin sulfate from the sea cucumber *Pentacta pygmaea* (PpFucCS) in order to block human cytomegalovirus (HCMV) entry into cells. Human foreskin fibroblasts were infected with HCMV in the presence of PpFucCS and its low molecular weight (LMW) fractions and the virus yield at five days post-infection was assessed. The virus attachment and entry into the cells were visualized by labeling the purified virus particles with a self-quenching fluorophore octadecyl rhodamine B (R18). The native PpFucCS exhibited potent inhibitory activity against HCMV specifically blocking virus entry into the cell and the inhibitory activities of the LMW PpFucCS derivatives were proportional to their chain lengths. PpFucCS and the derived oligosaccharides did not exhibit any significant cytotoxicity; moreover, they protected the infected cells from virus-induced lytic cell death. In conclusion, PpFucCS inhibits the entry of HCMV into cells and the high MW of this carbohydrate is a key structural element to achieve the maximal anti-viral effect. This new marine sulfated glycan can be developed into a potential prophylactic and therapeutic antiviral agent against HCMV infection.

## 1. Introduction

Betaherpesviruses, including human cytomegalovirus (HCMV), are common human pathogens, yet only a small percentage of infected people experience acute illness [1]. In people with impaired immune systems, HCMV-mediated infections can be life-threatening. Despite significant advancements in diagnostic and therapeutic management of the disease, HCMV poses a serious threat in solid organ and hematopoietic stem cell transplant recipients as well as in acquired immunodeficiency syndrome patients [2]. In addition, HCMV infection in utero is the leading cause of congenital infection that can result in developmental abnormalities, including sensorineural hearing loss, developmental delays, and even fetal death [3,4,5].

There are currently no commercially available vaccines to prevent HCMV infection [6] and only a small number of drugs, including ganciclovir, valganciclovir, cidofovir, foscarnet, maribavir, and letermovir, have been Food and Drug Administration approved for the treatment/prophylaxis of HCMV infection and disease [7,8]. Moreover, drug resistance is increasingly common with circulating strains acquiring UL54 (polymerase) and/or UL97 (kinase) mutations that confer resistance to commonly available drugs [9,10,11]. Therefore, the identification of newer anti-HCMV drugs with improved efficacy and novel modes of action are essentially needed.

Heparan sulfate (HS) is present on the cell surface as well as in the extracellular matrix of most mammalian tissues [12,13]. Numerous viruses including HCMV attach to host cells by first engaging with cell surface HS glycosaminoglycan chains, followed by interaction with the specific protein receptors, which results in virus entry [14,15]. HCMV can infect different types of cells and tissues indicating that it can engage with a wide variety of host cell receptors to enter the cell and establish an infection [16,17,18,19]. Specific viral glycoproteins engage these structurally and functionally distinct receptors that include the platelet-derived growth factor receptor alpha, cellular integrins, neuropilin-2, and epidermal growth factor receptor [20,21,22,23,24,25,26].

Earlier, our group showed that HCMV preferentially binds to uniquely sulfated and polymerized HS using its surface glycoprotein B (gB) [15]. HS mimetics, including heparin, bind to gB and inhibit HCMV infection by competitively inhibiting virus attachment to the cell surface [15,27]. Heparin is composed of disaccharide-repeating units (Figure 1A) and is an efficient anticoagulant; it can cause excessive bleeding and thrombocytopenia that necessitate close patient observation and, if necessary, the administration of an antidote (protamine) [28,29]. Two potential solutions to these anticoagulant side effects are using either the naturally potent anti-viral sulfated glycans with lower anticoagulant properties or the dissociation of the anticoagulant and anti-viral properties of the existing heparin. We were able to provide instances of both in our earlier work [28,30,31]. In contrast to heparin, a newly discovered fucosylated chondroitin sulfate (FucCS) from the body wall of the sea cucumber *Pentacta pygmaea* (PpFucCS) was found to have significantly lower anticoagulant activity but significant antiviral efficacy against severe acute respiratory syndrome coronavirus 2 (SARS-CoV-2) [28]. The PpFucCS anticoagulant activity can be further reduced by hydrolysis into oligosaccharide fractions [30].

In the current investigation, PpFucCS (Figure 1B) was used to target HS–virus interaction and prevent HCMV entry into cells. We show that PpFucCS has significant antiviral activity against HCMV and that, when hydrolyzed into fractions, the oligosaccharides lose their anti-HCMV activity in a size-dependent manner.

## 2. Materials and Methods

**PpFucCS extraction and purification:** As previously reported [30], PpFucCS was recovered from the body wall of the sea cucumber *P. pygmaea* (Gulf Specimen Lab Gulf of Mexico, Florida Keys) by performing proteolytic digestion using papain (Sigma, St. Louis, MO, USA). Using anion exchange chromatography, the dry crude extract was fractionated on a 1.5 cm × 20 cm column packed with DEAE-Sephacel resin (Sigma). It was then eluted using a linear gradient of NaCl in 0.1 M NaOAc at pH 6.0 from 0 M to 3 M with a flow rate of 0.5 mL/min. 1,9-dimethyl methylene blue (DMB) reagent was used to monitor each fraction. The polysaccharide fractions that were detected using the DMB assay were combined, dialyzed three times against water, and then lyophilized to concentrate for preservation and transportation. The dialyzed sugars were then subjected to purification using a Sephadex G15 (Sigma) column (1 cm × 30 cm) with water as the mobile phase for cleanup. The lyophilization was performed to purify the sugar. To estimate the concentration of the samples in the various assays, dry weight measurements were collected.

**PpFucCS Depolymerization:** PpFucCS was depolymerized using a modified Fenton technique that has been demonstrated to be extremely selective for the breakage of GlcA and GalNAc bonds found in the backbone of this class of polysaccharides [30]. A previously described methodology that had been slightly changed was used to carry out the reaction; 40 mg of dry-weight PpFucCS was weighed and dissolved at a concentration of 2 mg/mL in 0.1 M sodium acetate at a pH of 6.0 to optimize the reaction conditions; 0.02 mM copper (II) acetate (final concentration) was then added to the dissolved polysaccharide solution. The reaction was carried out for 180 min at 60 °C with constant stirring after the addition of 200 mM H_2_O_2_ in drops to the reaction liquid. By removing the copper ions from the reaction mixture using Chelex (50–100 mesh size) resin, the reaction was quenched by adding Chelex resin that had been pre-equilibrated using 0.1 M sodium acetate buffer, pH 6.0, and keeping it on an end-to-end rotor for 2 h at room temperature. The suspension was centrifuged at 3000 rpm for 10 min after 2 h. The supernatant was lyophilized and desalted.

**Fractionation of PpFucCS oligosaccharides:** The dried depolymerized mixture of PpFucCS was fractionated depending on the size in a Bio-Gel P-10 column (1.5 cm × 170 cm), at a flow rate of 1.0 mL/15 min. A 10% ethanol solution containing 1.0 M NaCl was used to fractionate the mixture. Hydrolyzed PpFucCS (HdPpFucCS) is native PpFucCS that has been subjected to depolymerization The fractions obtained were assayed for the presence of PpFucCS oligosaccharides using DMB. The eluted oligosaccharides were divided into 4 fractions (Fr1–Fr4) based on the DMB profile and their corresponding retention times on the Bio-Gel P-10 column. Before any analysis, all the fractions were desalted on the Sephadex G-15 column and then lyophilized. See [30].

**Distribution of PpFucCS oligosaccharides:** Polyacrylamide gel electrophoresis (PAGE) was used to analyze the oligosaccharide distribution of the depolymerized PpFucCS fractions [30].

**PpFucCS oligosaccharides structural integrity determination:** The structural integrity of the PpFucCS oligosaccharides produced was investigated using one-dimensional proton nuclear magnetic resonance (1D ^1^H NMR) spectroscopy [30].

**Cells:** The human foreskin fibroblasts (HFF) were cultured in Dulbecco’s modified Eagle’s medium (DMEM) (Corning, Manassas, VA, USA, catalog No. 10-013-CM) with 10% fetal bovine serum (FBS) (Gibco, Life Technologies, Grand Island, NY, USA, catalog No. 10437-028), 2 mM L-glutamine, and 100 U/mL penicillin-streptomycin (Corning, Manassas, VA, USA; catalog No. 30-002-CI) at 37 °C along with 5% CO_2_ [32].

**Virus:** HCMV (TowneBAC strain) tagged with a green fluorescent protein (GFP) [33] was grown on the HFF cells. The virus stock was prepared in 3X autoclaved milk, sonicated three times for 10 s with a 30 s gap, and stored at −80 °C. For the preparation of the autoclaved milk, we dissolved Nestle Carnation instant nonfat dry milk powder in nanopure water at neutral pH to obtain 10% milk [6,15,34], which was then autoclaved three consecutive times before storage at 4 °C.

**Virus infections:** For infecting the HFFs, the media was removed from the wells of the cell culture plates, washed with serum-free medium DMEM, and appropriately diluted virus stock was absorbed on the cells in DMEM without serum. The cells were incubated for 1 h with gentle shaking every 10 min followed by washing thrice with serum-free DMEM. Fresh complete DMEM was added, and the cells were incubated until the endpoint.

**IC_50_ assay and percent inhibition calculation:** To determine the half-maximal inhibitory concentration (IC_50_) of PpFucCS, its hydrolyzed form, and its fractions, the HFFs were pretreated for 1 h with a range of concentrations (50 μg/mL, 25 μg/mL, 12.5 μg/mL, 6.25 μg/mL, 3.12 μg/mL, 1.56 μg/mL, and 0.78 μg/mL), UFH, or mock, and then infected with HCMV at a multiplicity of infection (MOI) of 0.1. The cells were fixed at 5 days post-infection (dpi), and the number of GFP-positive cells for each concentration was enumerated using the Cytation 5 automated microscope (BioTek Instruments, Inc., Agilent, Santa Clara, CA, USA). The experiment was repeated three times in order to confirm the reproducibility of the measurements. The data were normalized in GraphPad Prism 9.0, using the number of GFP-positive cells in the mock wells as 0% inhibition and the number ‘0′ as 100% inhibition. The percentage (%) inhibition was plotted against the concentration range. The relative IC_50_s were calculated using a fixed top limit of the average vehicle-only control and a floating bottom limit.

For virus yield reduction assays, the HFFs were pretreated for 1 h with a range of concentrations (50 μg/mL, 25 μg/mL, 12.5 μg/mL, 6.25 μg/mL, 3.12 μg/mL, 1.56 μg/mL, and 0.78 μg/mL), of PpFucCS and mock (in duplicates of two independent experiments), and then infected with HCMV at a MOI of 3. The samples were harvested within the medium at 5 dpi by scraping the cells and stored at −80 °C before titration. On the day of titration, the harvested samples were sonicated three times for 10 s each with a 30 s gap. The monolayers of the HFFs were grown in 12-well tissue culture plates, and serial dilutions of the sonicated samples in the DMEM (without any additional compounds) were absorbed onto them for 1 h (in duplicates), followed by washing thrice with serum-free DMEM. For the viral titers, fresh DMEM containing 10% FBS and gamma immunoglobulin was added to the HFFs. At this step, no native PpFucCS was added to the overlayed medium, and the cells were incubated for from 9 to 10 dpi. At the endpoint, the DMEM overlay was removed, and the cells were washed 2X using PBS. The infected HFFs were then fixed in 100% methanol for 5 min. The HFFs were immediately stained using Modified Giemsa stain (Sigma-Aldrich, Milli-pore Sigma, Burlington, MA, USA, catalog No. GS1L) for 15 minutes. the plates were then washed with tap water and air dried, and the dark stained plaques were quantified.

**Octadecyl rhodamine B (R18) labeling and entry assay:** The appropriate amount of virus was incubated using R18 (20 ng/L) (Thermo Fisher Scientific, Invitrogen, Waltham, Massachusetts, USA, catalog No. O246) at 4 °C for 1 h. An equal volume of DMEM was added and the tube was centrifuged at 28,000 rpm for 1 h at 4 °C to pellet the labeled virus, which was resuspended in 200 μL of DMEM. The native PpFucCS (50 μg/mL) was overlayed on the cells (HFF) for 1 h at 37 °C, and the plate was then incubated at 4 °C for 15 min. The purified virus, labeled with R18, was added to the plate and incubated for 1 h at 4 °C with the pre-treated cells to allow for virus attachment. Thereafter, the unbound virus was washed off using PBS and the plate was moved to 37 °C to allow for virus entry. The plates were fixed for staining with Hoechst dye at different time points after the 37 °C move. The mock-treated cells with the labeled and unlabeled virus were used as controls.

**Microscopy:** The samples were prepared using the established protocols for fluorescence microscopy. Briefly, the HFF cells were grown on coverslip inserts in 24-well tissue culture plates. The cells were pretreated using PpFucCS at 50 μg/mL concentration for 1 h and cooled at 4 °C for 15 min before infection using R18-labeled HCMV at a MOI of 3.0. The plate was then incubated at 37 °C along with 5% CO2 for 1 h. At the different time points (0, 1, 2, 5, 10, 20, 40, and 60 min) after the shift to 37 °C, the cells were fixed in 3.7% formaldehyde for 10 min and incubated in 50 mM NH_4_Cl in phosphate-buffered saline (PBS) for 10 min to reduce autofluorescence and washed using 2X PBS. Finally, the cells were incubated in Hoechst solution (ThermoFisher Scientific, Waltham, MA, USA, Catalog No. 33342) in PBS (1:3000) for 10 min to stain the nucleus, followed by a PBS wash. The coverslips were retrieved from the wells and were mounted on glass slides with a drop of mounting medium (2.5% DABCO in Fluoromont G) and air dried for two hours before imaging. The images were acquired on an EVOS-FL epifluorescent microscope (ThermoFisher Scientific, Waltham, MA, USA).

**Cell viability:** The HFFs were plated in 24-well tissue culture plates and grown to confluency (in triplicates). The cells were pretreated for 1 h with the highest concentration of test compounds and controls and then infected with HCMV at a MOI of 3.0 or mock infected. The medium was removed at 5 dpi, and the HFFs were harvested using trypsinization. The cell viability was determined using trypan blue exclusion using a TC20 automated cell counter (BioRad Laboratories, Hercules, CA, USA) following the manufacturer’s protocol. The test was also performed using the native PpFucCS at 500 µg/mL via the trypan blue exclusion assay along with the ATP release assay. We performed the bioluminescent ATP assay by overlaying the compounds for five days on the confluent HFF in an opaque-walled 96-well tissue culture plate with blank controls using a CellTiter-Glo 2.0 Cell Viability Assay kit (Promega Corporation, Madison, WI, USA, Catalog No. G9242) according to the manufacturer’s protocol. The ATP luminescence was measured using a Cytation 7 automated microscope (BioTek Instruments, Inc.) with the Endpoint/Kinetic read type using a single wavelength with the appropriate gain. The viability experiments were repeated two times to confirm the measurements.

**Statistics:** The data were plotted in Graphpad Prism (Version 9.0, GraphPad Software, San Diego, CA, USA, www.graphpad.com, accessed on 3 March 2023). The standard error of the mean was plotted as error bars. A *p*-value of less than 0.05 was considered significant. An asterisk (*) denotes a significant inhibition compared to the mock. The data were evaluated using one-way ANOVA in GraphPad Prism 9.0 for multiple comparisons, and the differences between the groups were considered significant at a *p*-value of less than 0.05.

## 3. Results

### 3.1. Inhibition of HCMV by PpFucCS and Its Fractions

Here, we investigated the inhibitory activity of both native PpFucCS and its LMW fractions against HCMV. The LMW PpFucCS oligosaccharides studied here were used to establish a structural activity relationship and the influence of MW distribution on its anti-viral activity against HCMV.

The native PpFucCS exhibited potent and efficacious inhibitory activity against HCMV while the PpFucCS fractions showed decreased activity in both potency and efficacy (Figure 2A). Comparing the mean inhibition at the highest concentration used (50 μg/mL), we show that the efficacy of native PpFucCS along with the unfractionated but hydrolyzed PpFucCS oligosaccharide mixture (HdPpFucCS) and Fr1 was comparable to UFH, but it was significantly lower in the case of Fr2, Fr3, and Fr4 (Figure 2B). Additionally, the native PpFucCS had the least half maximal inhibitory concentration (IC_50_) value amongst all the tests and was comparable to that of UFH (Table 1). Thus, based on the comparison of low MW PpFucCS products, it is evident that HdPpFucCS and Fr1 maintain better antiviral activity than Fr2, Fr3, and Fr4 while displaying only slightly less activity than native PpFucCS and UFH. Earlier, our 1D ^1^H NMR spectra showed the presence of four fucose units in Fr1–Fr3 that were similar to the native PpFucCS structure: α- Fuc2,4S, α-Fuc2,4S, α-Fuc4S, and α-Fuc0S [30]. This indicates that, although Fr1–Fr3 differ in size, they have similar structures. The Fuc anomeric peaks in fraction 4 showed an upfield ^1^H shift, which may indicate desulfation. Interestingly, the Fenton chemistry demonstrated a preserved structural integrity of the native PpFucCS in the majority of the oligosaccharides produced (Fr1–Fr3), and only a small percentage of the very low MW (Fr4) showed more chemical modifications [30]. Combining these recently published results with the current results, it is evident that PpFucCS oligosaccharides lose their anti-HCMV activity in a size-dependent manner and that structural features play a crucial role in their anti-viral activity.

To study the inhibition of progeny virion formation and virus growth, we performed a virus yield reduction assay (at MOI 3.0). The IC_50_ obtained by performing this assay (2.957 µg/mL) (Appendix A) is very close to the IC_50_ (3.6 µg/mL) determined by the GFP reduction assay conducted at a low MOI (0.01) (Table 1), thus indicating that the inhibitory properties of PpFucCS against HCMV do not depend on the MOI.

### 3.2. PpFucCS Specifically Inhibits HCMV Entry into Cells

To assess whether PpFucCS inhibits the entry of HCMV into cells (HFFs), we utilized the TowneBAC strain of the HCMV labeled with a self-quenching fluorescent label (R18 fluorescence assay). The fluorescence was recorded at different time intervals starting from 0 to 60 min post infection (Figure 3). The R18 dye transitions from one monolayer of a fluid-state phospholipid bilayer membrane to another in a potential-dependent manner, somewhat relieving the quenching of its fluorescence [35]. The fluorescence from the HCMV virus particles that have been labeled with high quantities of R18 is highly self-quenched and their fusion with the host cell membrane leads to an increase in fluorescence, which is often visible only after the virus has entered the cell. A significantly higher number of virus particles entered the mock-treated cells compared to the cells treated with native PpFucCS at 60 min post infection (Figure 3A,B).

### 3.3. Effect of PpFucCS and Its Oligosaccharides Treatment on Cell Viability

PpFucCS and its oligosaccharides do not exhibit any significant cytotoxicity against primary HFF cells when examined at the highest concentration used in the antiviral assay (50 μg/mL) (Figure 4A) and show a protective effect against virus-induced cell death at five days post infection (Figure 4B) when examined using a trypan blue exclusion assay. Notably, the Fr3 and Fr4 that did not show any significant antiviral activity (Figure 2) also did not show any significant protection from virus-induced cell death. Furthermore, we did not see any cytotoxic effect on the HFFs when treated with the native PpFucCS at 500 μg/mL for five days using the trypan blue exclusion assay (Appendix A). To corroborate these findings, we treated the HFFs with the native PpFucCS at 500 μg/mL and 50 μg/mL along with heparin control for five days and performed the ATP release assay (Appendix A).

## 4. Discussion

In this study, we demonstrated that PpFucCS and its hydrolyzed fractions efficiently inhibit the HCMV infection of primary human fibroblasts, specifically blocking virus entry into the cells. The treatment of HFFs with native PpFucCS and its oligosaccharides did not affect the cell viability for the duration of treatment, thus confirming that the observed reduction in the virus titer was not because of cytotoxicity. Moreover, when the HFFs were pretreated and maintained in these marine glycans throughout the course of infection, the cells resisted infection-induced cell death. To confirm that PpFucCS has a direct impact on virus entry, we used HCMV labeled with self-quenching dye that only fluoresces when the virus fuses with the cell membrane. This experiment showed the significant effect of PpFucCS on HCMV entry.

GAGs are composed of repeating disaccharide units that contain an amino sugar (either GalNAc or GlcNAc) and a uronic acid (either IdoA or GlcA) or the neutral sugar galactose. GAGs have a variety of biological functions, including cell signaling, proliferation, and wound healing, as a result of their structural diversity, and are ubiquitously found on the surface of cells, where they act as receptors and signals for cellular and pathogenic processes [36]. Viruses or other pathogens utilize this structural diversity of GAGs to selectively bind and enter the cells [37]. The virion attachment caused by GAG binding and eventual entry are usually independent events, with attachment assumed to be mostly nonspecific and charge-dependent but, instead, entry is thought to be driven by highly specific protein–protein interactions. We have recently shown that HS–virus interactions are specific and depend on a defined structural functional relationship [12,15,28]. Interactions with HS are a frequent prerequisite for many viral infections [38,39]. As a result, inhibitors targeting virus–HS interactions have the potential to function as broad-spectrum antivirals that could have a significant influence on global health.

Many compounds including small organic and inorganic compounds, as well as polycationic peptides, have been employed to bind to HS and thereby protect the cells from viral attachment [40,41,42,43]. Conversely, inhibitors may bind to virion components to sequester and inactivate the virions very similarly to neutralizing antibodies [44]. These are mostly HS mimetics, and many of them have broad-spectrum antiviral action. In this study, we investigated the anti-HMCV activities of two GAGs: (i) heparin, a highly sulfated linear GAG widely exploited as in research [45], and (ii) PpFucCS, a unique marine branched GAG. From our results, we were able to compare the anti-HCMV activity of PpFucCS and its LWW fractions with the commonly used control heparin. Since the use of heparin can lead to excessive bleeding and heparin-induced thrombocytopenia, PpFucCS offers a reasonable alternative to be developed as an antiviral.

The inhibitory activities of the LMW PpFucCS derivatives were proportional to their chain lengths. The native high MW structure is necessary to achieve maximum HMCV inhibition. Upon hydrolysis (depolymerization) to generate the oligosaccharide mixture (HdPpFuCS), the activity was considerably decreased. However, Fr1 had more anti-HMCV activity compared to HdPpFucCS, most likely because HdPpFucCS has significant amounts of less active components (Fr2–Fr4: nearly 3/4 of the amount). Moreover, Fr4 has a degraded structure. All the PpFucCS fractions (Fr1–Fr3) show the same 1D 1H NMR spectral profile as the native, therefore indicating they have the same structure [30]. The only exception is Fr4, which was spectrally different and structurally degraded.

We recently observed a similar behavior of MW dependence on anticoagulation and anti-SARS-CoV-2 actions of PpFucCS [30]. However, for the sulfated galactan from the red alga *Botryocladia occidentalis* (BoSG), the reduction in the chain length was able to dissociate the anticoagulant and anti-SARS-CoV-2 activities [31]. The results from this current work support the first case, in which the high MW of the marine sulfated glycan, in this case PpFucCS, is a key element for the biological activity. The current results regarding the anti-HCMV activity of PpFucCS perfectly match our recent findings regarding the anticoagulation and anti-SARS-CoV-2 action of PpFucCS [30].

## 5. Conclusions

The current results demonstrate the antiviral activities of marine sulfated glycan PpFucCS against HCMV, which is an enveloped DNA virus. Earlier, our group successfully used a copper-based Fenton approach for the free-radical depolymerization of PpFucCS to produce oligosaccharides with shorter chain lengths but preserved the structural integrity, so it was similar to the native PpFucCS [30]. Our findings imply that a native PpFucCS structure with a large MW is essential for achieving the greatest anti-HCMV activity. The retained effective inhibitory potential of high MW structures derived from PpFucCS suggests this marine glycan is a promising candidate for further evaluation and development as a potential antiviral.

## Figures and Tables

**Figure 1 viruses-15-00859-f001:**
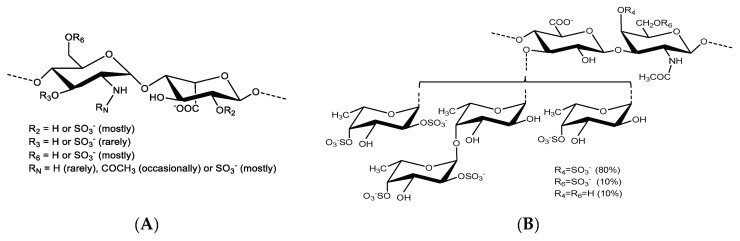
Structural representation of sulfated glycans assayed for anti-HCMV activity. (**A**) Unfractionated heparin (UFH) is mostly composed of repeating disaccharide units of [→4)-α-GlcN-(1→4)-α-IdoA-(1→] where GlcN is glucosamine and IdoA is iduronic acid. Sulfation occurs frequently at the N- and C6-positions of GlcN and C2 position of IdoA. (**B**) PpFucCS is constituted of a chondroitin sulfate backbone that alternates N-acetylgalactosamine (GalNAc) and glucuronic acid (GlcA) in repeating disaccharide units of [→3)-β-GalNAc-(1→4)-β-GlcA-(1→], where the GalNAc units are primarily 4-sulfated (80%) and, to a very less degree, 4,6-disulfated (10%) or nonsulfated (10%). The GlcA units are replaced at the C3 position by three different forms of α-fucose (Fuc) branches: Fuc2,4S-(1→(40%), Fuc2,4S-(1→4)-Fuc-(1→(30%), and Fuc4S-(1→(30%); where S = SO_3_^−^.

**Figure 2 viruses-15-00859-f002:**
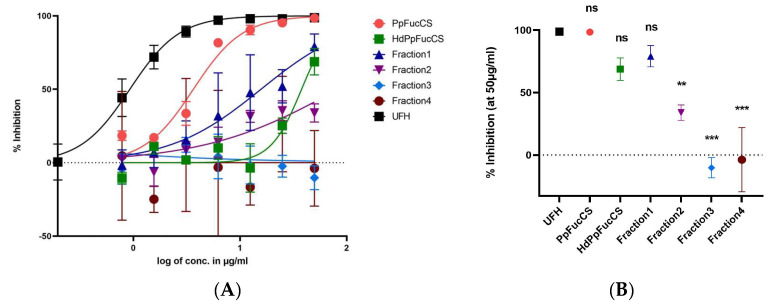
Anti-HCMV activity of PpFucCS oligosaccharides. PpFucCS oligosaccharides were assayed for their potential to inhibit the HCMV (GFP-tagged TowneBAC strain) infection in HFFs (MOI 0.1) by enumerating the cells expressing GFP. (**A**) Normalized values from the assay were analyzed by nonlinear regression to fit a dose–response curve using the least squares method considering each repeated measure as an individual point. The plotted curve shows the percentage of HCMV inhibition in a (log) concentration-dependent manner. Curves in the plot represent the following: PpFucCS (red), HdPpFucCS (green), Fr1 (navy), Fr2 (purple), Fr3 (blue), Fr4 (brown), and UFH (black). (**B**) Using the normalized values from the same assay, we calculated the percentage inhibition at the maximum concentration used (Imax) and compared with the UFH at the same concentration (50 µg/mL) by performing one-way ANOVA test with multiple comparisons by comparing the means of each test with control UFH, and corrected using a Dunnett’s post hoc test, showing significant differences among the means. The results are representative of three independent experiments. The standard error of the mean was plotted as error bars. The ***, **, and ns (non-significant) indicate *p*-value < 0.001, from 0.001 to 0.01, and ≥0.05, respectively.

**Figure 3 viruses-15-00859-f003:**
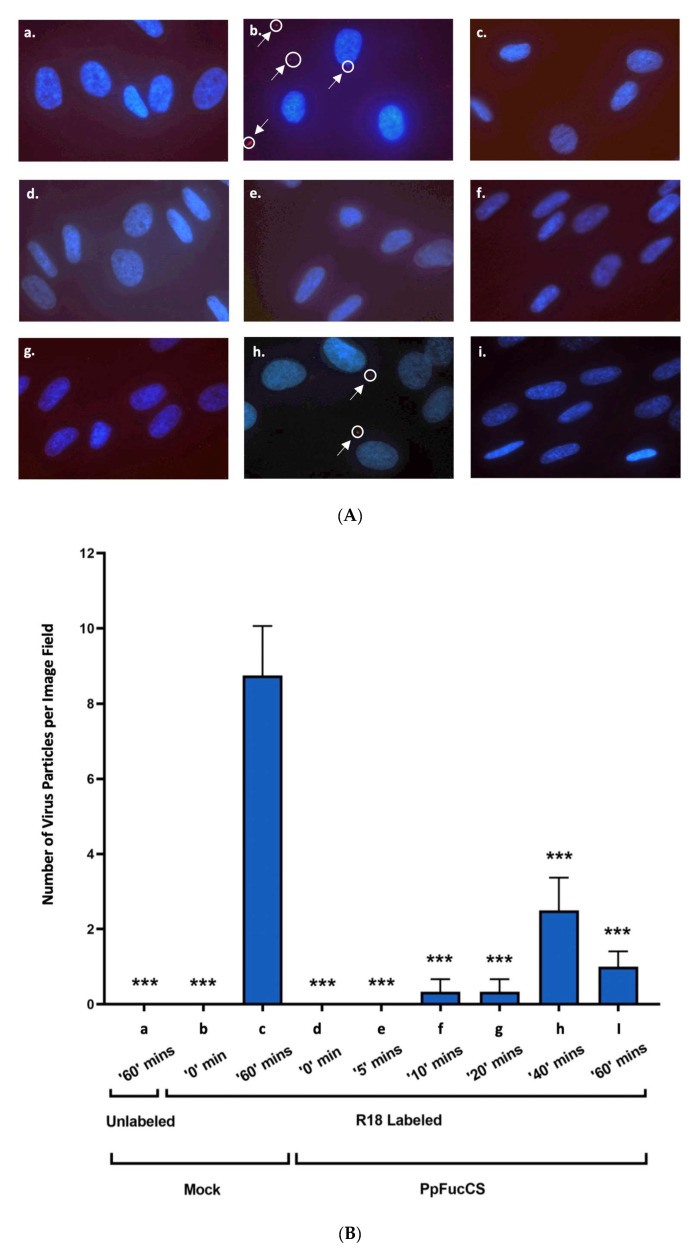
Inhibition of HCMV entry into the HFF cells by native PpFucCS. The primary HFF were challenged by the unlabeled (**a**) or R18 labeled (**b**–**i**) TowneBAC strain of HCMV at a MOI of 3.0. The HFFs were either mock treated (**a**–**c**) or pretreated for one hour using PpFucCS (50 μg/mL) (**d**–**i**). The number of fluorescent virus particles were enumerated at different time points post infection and plotted in Graphpad Prism 9. The HFFs were either mock treated and infected with the unlabeled virus at ‘60′ min (**a**), or mock treated and infected with the R18 labeled virus at ‘0′ min (**b**), ‘60′ min (**c**), or PpFucCS-treated and infected with the R18 labeled virus at ‘0′ min (**d**), ‘5′ min (**e**), ‘10′ min (**f**), ‘20′ min (**g**), ‘40′ min (**h**), and ‘60′ min (**i**). (**A**) Images showing the different groups either treated or mock treated. The cell nuclei appear blue fluorescent while the red fluorescent virus particles are indicated by the circle and arrows. (**B**) Bar plot represents the number of fluorescent virus particles per image field for treatment groups at different time points. The results are representative of four independent experiments and were analyzed by performing one-way ANOVA test with multiple comparisons (comparing the means of each test with mock-treated control at ‘60′ min). The standard error of the mean was plotted as error bars. The ***, and ns indicate *p*-value < 0.001, and ≥0.05, respectively.

**Figure 4 viruses-15-00859-f004:**
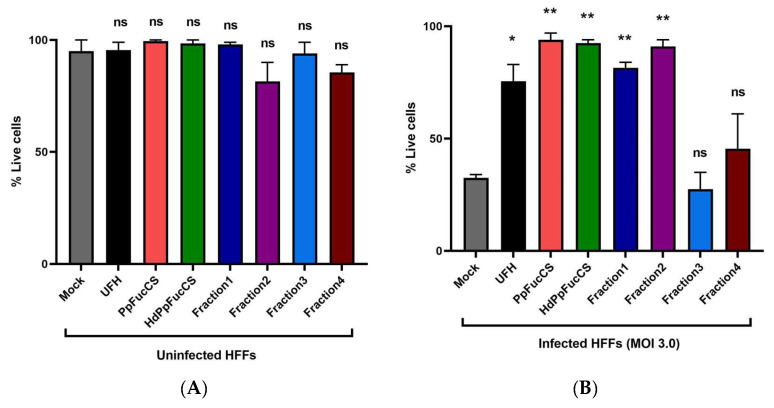
Effect of treatment of PpFucCS and its oligosaccharides on cell viability of HFF cells. The primary HFF were pretreated for one hour using PpFucCS and its oligosaccharides at the highest concentration used for inhibition assays (50 μg/mL) along with heparin (UFH) control. The HFFs were either mock infected (**A**) or infected with HCMV at a MOI of 3.0 (**B**) in the presence of test compounds. Cells were harvested at 5 days post-infection and cell viability was assessed using a trypan blue exclusion assay. The results are representative of two independent experiments. The standard error of the mean was plotted as error bars. The **, *, and ns indicate *p*-value between 0.001 and 0.01, between 0.01 and 0.05, and ≥0.05, respectively.

**Table 1 viruses-15-00859-t001:** IC_50_ values of anti-viral activities of PpFucCS and its oligosaccharides. The IC_50_ values of anti-HCMV inhibitory activity of PpFucCS and its oligosaccharides were determined in HFFs. Considering the role of avidity often found in protein–glycosaminoglycan (GAG) interactions and the broad range of molecular weights of the compounds, IC_50_ values were measured in terms of µg/mL rather than the molar units. ND, not detected.

Compounds	IC_50_ (µg/mL)	SEM (µg/mL)	Lower 95% Conf. Limit (µg/mL)	Upper 95% Conf. Limit (µg/mL)
**PpFucCS**	3.6	0.3	3.0	4.2
**HdPpFucCS**	37.8	4.7	27.7	47.9
**Fraction1**	15.7	5.4	4.2	27.3
**Fraction2**	83.1	39.9	0	166.7
**Fraction3**	ND	ND	ND	ND
**Fraction4**	ND	ND	ND	ND
**UFH**	0.9	1.946	0	4.9

## Data Availability

All data are included in the manuscript itself.

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
