# Peer review of "Inhibition of Cytomegalovirus by Pentacta pygmaea Fucosylated Chondroitin Sulfate Depends on Its Molecular Weight"

_viruses, 2023, doi:10.3390/v15040859_

Round 1

Reviewer 1 Report

The manuscript by Sharma et al. deals with the antiviral effects of fucosylated chondroitin sulfate against human cytomegalovirus. In this manuscript the authors extracted FucCS  from the body wall of the sea cucumber P. pygmaea as well as hydrolysis into oligosaccharide fraction, although this has been described by the authors earlier (Dwivedi et al., 2021,2022). Further, the authors performed analysis to determine the efficacy on viral entry and cell viability. The authors demonstrated that the native structure with large MW is a prerequisite for achieving the highest antiviral effects. Overall, the manuscript describe promising antiviral effects of the PpFucCS, thus leading to a potential candidate for new antiviral development. However, not every statement is appropriate illustrated in the experiments. After addressing the critical points, the manuscript is suitable for publication in Viruses.

Major points

1.       Virus source: The authors should either include a reference of the construction of TowneBac-GFP or a complete description.

2.       What is the reason to store the virus stock in 10% milk? This is very unusual and not used by the community.

3.       In many experiments, it is described that replicas were used. In general, experiments have to be done at least in duplicates and three independent experiments. The author should be aware of this point.

4.       Determination of the EC50 is more reliable when plaque reduction assays were used as the read out. This has to be done before publication. In addition, since the authors knew the MW of all compounds they should use µM instead of µg/ml.

The MOI is very high compared to the literature. The authors should show the effects with two lower MOIs.

5.       Determination of the CC50 is missing. To verify whether the compounds were biologic active molecules the selectivity index has to be determined. SI is the value of CC50/EC50 and should be above 10.

6.       Regarding the analysis of viral entry the authors should use an overlay with a temperature shift after virus inoculation and plaque assays at appropriate time points. The results will lead to a more reliable quantification of the effects.

However, the authors could use their technique but need high-resolution imaging to determine the localization of substances on the cell surface and to determine if attachment or penetration is inhibited. Therefore, superresolution microscopy using structured illumination should be carried out.

These analysis have to be done before acceptance.

Author Response

Review report 1

  1. Virus source: The authors should either include a reference of the construction of TowneBac-GFP or a complete description.

Thank you for your advice. The construction of TowneBac-GFP is mentioned in the following article. We have added reference no. 33 to the reference list. See addition at pg 3, ln 135.

Dunn W, Chou C, Li H, Hai R, Patterson D, Stolc V, Zhu H, Liu F. Functional profiling of a human cytomegalovirus genome. Proc Natl Acad Sci U S A. 2003 Nov 25;100(24):14223-8. doi: 10.1073/pnas.2334032100. Epub 2003 Nov 17. PMID: 14623981; PMCID: PMC283573.

  1. What is the reason to store the virus stock in 10% milk? This is very unusual and not used by the community.

In a herpesvirus laboratory, it is common practice to store the virus stock in 10% milk. This preserves the integrity of virus stocks during freeze thaws. We are listing refence no. 6, 15 and 34 on the line no. 139, pg 4, to provide precedents of this practice.

  1. In many experiments, it is described that replicas were used. In general, experiments have to be done at least in duplicates and three independent experiments. The author should be aware of this point.

We appreciate the reviewer’s point. We have changed the language at several places in the text to convey that the experiments were repeated for reproducibility and at least a duplicate sample was used for each experimental run. See added comments at pg 4, ln 158-172.

  1. a) Determination of the EC50 is more reliable when plaque reduction assays were used as the read out. This has to be done before publication.

As suggested by the reviewer, we have now included the results of a plaque reduction assay testing the inhibitory activity of native PpFucCS. The results of these findings are displayed in fig S1 and confirm the earlier results obtained using GFP reduction assays.

  1. b) In addition, since the authors knew the MW of all compounds, they should use µM instead of µg/ml. The exact molecular weight of the PpFucCS and derivates is difficult to determine. In an earlier publication, https://doi.org/10.1093/glycob/cwac063” in figure 1. B) our PAGE analysis showed that the molecular weights of these sugars have a broad range, which makes it difficult to determine IC50 in molar units. Also, IC50values measured in terms of µg/ml give a better representation of the inhibitory activity since molecular avidity can often influence protein-GAG interactions.

  1. c) The MOI is very high compared to the literature. The authors should show the effects with two lower MOls.

We have now performed plaques assays at a high MOI of 3 using native PpFucCS and determined an IC50 (2.957 µg/ml) that is very close to the IC50 (3.6 µg/ml) determined by GFP reduction assay conducted at a low MOI (0.01) (fig S1) indicating that the inhibitory properties of PpFucCS against HCMV do not depend on MOI.

  1. Determination of the CC50 is missing. To verify whether the compounds were biologic active molecules the selectivity index has to be determined. SI is the value of CCso/ECso and should be above 10.

We have conducted an additional experiment- trypan blue exclusion assay to show that our compound native PpFucCS does not show any cytotoxicity even at a very high concentration of 500 μg/ml. This makes it very difficult to determine the CC50 since it would require much higher quantities of PpFucCS to induce any noticeable cell death, and these higher quantities are technically difficult to obtain. See additions on pg 5, ln 203-212, and pg 8, ln 343 and 345-349.

  1. a) Regarding the analysis of viral entry the authors should use an overlay with a temperature shift after virus inoculation and plaque assays at appropriate time points. The results will lead to a more reliable quantification of the effects.

We have now included the results of our plaque assays (fig S1). The R18 labeling experiments used temperature shifting where virus was allowed to accumulate at the cell surface at 4 oC and thereafter shifted to 37 oC to allow for virus entry.

  1. b) However, the authors could use their technique but need high­ resolution imaging to determine the localization of substances on the cell surface and to determine if attachment or penetration is inhibited. Therefore, super-resolution microscopy using structured illumination should be carried out.

Currently, the super-resolution microscopy with structured illumination is not available in our laboratory.  We have used a fluorescent microscopic imaging at 100X magnification which is sufficient to track fluorescence emanating from individual virus particles once they enter the cells.

Reviewer 2 Report

The manuscript titled “Inhibition of Cytomegalovirus by Pentacta pygmaea Fucosylated Chondroitin Sulfate Depends on its Molecular Weight” by Waters et al. investigated the antiviral properties of PpFucCS against HCMV. Many viruses attach to host cells via proteoglycans with HS glycosaminoglycan chains. PpFucCS inhibited HCMV entry into cells and correlated with its molecular weight. PpFucCS and its low molecular weight derivatives protected infected cells from virus-induced cell death without cytotoxicity. PpFucCS may be a promising HCMV prophylactic and therapeutic agent. I believe this study provides the valualbe information in this field.

The progress of the study is very intuitive and the background knowledge is well organized, but I hope that some additional supporting data will be added before publishing.

1.     Please show the effect of inhibiting progeny virion propagation by comparing CMV plaque size or GFP spreading size according to PpFucCS treatment.

2.     Could you draw virus growth curves following infection with a high MOI? At MOI 3, the entry of viral particles differs by 3- to 5-fold. It would be interesting to determine how much difference there is in progeny production.

3.     Consider combining Supplementary Figure 1 and Figure 3. In terms of convenience for readers, it would be nice to merge these two. How about the images of Supplementary Figure 1 are matched under the bar graph of Figure 3?

4.     Analyzing cytotoxicity using WST or MTT is more accurate than Trypan blue exclusion assay.

Author Response

Review report 2

  1. Please show the effect of inhibiting progeny virion propagation by comparing CMV plaque size or GFP spreading size according to PpFucCS treatment.

We have now performed the plaque assay using two-fold serial dilutions of native PpFucCS from 50 µg/ml and 0.78 µg/ml and we found similar sized plaques in both the control and treated groups. The data from this experiment is now included in supplementary fig S2.

  1. Could you draw virus growth curves following infection with a high MOI? At MOI 3, the entry of viral particles differs by 3- to 5-fold. It would be interesting to determine how much difference there is in progeny production.

Yes, we have repeated the inhibition curve at high MOI of 3 with plaque reduction assays and we have found a similar IC50 for native PpFucCS as compared to determining by fluorescent microscopy. See added comment at pg 6, ln 245-251.

  1. Consider combining Supplementary Figure 1 and Figure 3. In terms of convenience for readers, it would be nice to merge these two. How about the images of Supplementary Figure 1 are matched under the bar graph of Figure 3?

Thank you for your suggestion, we have included the data from supplementary figure 1 into figure 3 B in the manuscript.

  1. Analyzing cytotoxicity using WST or MTT is more accurate than Trypan blue exclusion assay.

To consider a different assay for cytotoxicity, we have now used ATP release assay for the treatment groups: native PpFucCS, heparin and mock. The figure has been included in the supplement (fig S4).

Reviewer 3 Report

In this manuscript Sharma et al describe a novel set of glycans (fucosylated chondroitin sulfates) from the sea cucumber Pentacta pygmaea that block CMV entry.  The isolated PpFucCS was then fractionated (as in their 2021 JBC paper) and depolymerized into the individual components and showed that the effect was lost in fractions 2-4 .   This is similar to their work with this glycan and SARS COV2 but now with a herpesvirus.  Both viruses rely on HS for entry. 

Major revisions:

1) Is this really blocking the HS interaction?  Compton et al (Virology 1993) showed that chondroitin sulfate does not block CMV entry so how is this glycan functioning?  This underlying mechanism is (potentially) the most interesting part of their findings.  They add the PpFucCS to the cells and NOT the virus.  If this is “mimicking” HS, it should bind to the virus and not necessarily to the cells.  If this is an HS “mimetic” (as it is used as a control in many of the experiments) why is it incubated with the cells and not the virus?      For example: the PpFucCS could bind to the cell and then lead to a down regulation of the other “attachment” proteins on the cells (i.e., EGFR, neuropilin, etc).  They need to convince the reader that it is acting on the virus and not the cell (or is it?).  How is this reconciled with the other findings?

2) They need to see if PpFucCS can block entry on other cell types that use other entry mechanisms/proteins.  In most cases the HS interaction is the first step, but it is unclear if these glycans are blocking the HS interaction.

3) The use of the R18 stained virus is not convincing.  This is the basis of much of their data.  Was the R18 stained virus/entry experiment done double blinded?  This seems pretty subjective and would merit more valid experimentation.  How many fields were enumerated?  This manuscript would benefit to have a more “traditional” HCMV entry experiment (i.e., FL-labeled glycoprotein in the virion then measure attachment at 4C).  It may be brighter than the R18 labeled virions.  Confocal microscopy to visualize?  Why not a traditional plaque reduction assay instead of relying on microscopy ?

4) Supplemental Figure: hard to tell whether the viruses are attached or not.  The counter stain is the nucleus, and it is hard to visualize the outer membrane where the virus should be attaching.  It is confusing why there is virus at 40 min and no virus at 60 min with treatment.  This relates to Fig. 3.  They need to have parallel time points with treated and untreated. 

5) The differences in the fractions are interesting and should be explored further and may indicate how these novel glycans are preventing infection.  Why does Fraction 1 and the depolymerized glycan still prevent attachment (Fig. 2) while 2-4 do not?

6) They say in line 217 that “structural features play an crucial role in” anti CMV activity but they also say that “Fr1-3 differ in size, they have similar structures” (line 210-211).  How do they reconcile this?   Maybe they are referring to sulfations?  If so, they need further experimentation to prove this.

Author Response

Review report 3

1) Is this really blocking the HS interaction? Compton et al (Virology 1993) showed that chondroitin sulfate does not block CMV entry so how is this glycan functioning? This underlying mechanism is (potentially) the most interesting part of their findings. They add the PpFucCS to the cells and NOT the virus. If this is “mimicking” HS, it should bind to the virus and not necessarily to the cells. If this is an HS “mimetic” (as it is used as a control in many of the experiments) why is it incubated with the cells and not the virus? For example: the PpFucCS could bind to the cell and then lead to a down regulation of the other “attachment” proteins on the cells (i.e., EGFR, neuropilin, etc). They need to convince the reader that it is acting on the virus and not the cell (or is it?). How is this reconciled with the other findings?

HCMV binding to cell surface glycosaminoglycans depends upon monosaccharide composition, sulfation levels, patterns of sulfation and polysaccharide chain length (PMID: 34352038).  As we have shown earlier, HCMV binds to large size chondroitin sulfate D, but not to chondroitin sulfate AC. It is important to note that while the chondroitin sulfate A (CS-A) is sulfated at C4 of the GalNAc, and the chondroitin sulfate C (CS-C) is sulfated at the C6 of the GalNAc only, the chondroitin sulfate D is sulfated at C2 of the glucuronic acid as well as the C6 of the GalNAc sugar and hence has double the amount of sulfation compared to CS-A and CS-C. Adding of HS mimic to the cell surface represents the conditions where these HS would be utilized for therapy in patients.  Across the Herpesviridae family, gB is one of the most conserved glycoproteins. For HCMV, our group has used purified HCMV gB and showed that it exhibits a very strong binding to heparin using surface plasmon resonance. We also found that UFH and it’s 6-desulfated form significantly competed with gB-bound heparin.

2) They need to see if PpFucCS can block entry on other cell types that use other entry mechanisms/proteins. In most cases the HS interaction is the first step, but it is unclear if these glycans are blocking the HS interaction.

The HSPGs are expressed on the cell surface of most mammalian cells.  In our earlier study (Ref #15) we have shown that fibroblast as well as endothelial cells show HS-dependent CMV entry.

3) The use of the R18 stained virus is not convincing. This is the basis of much of their data. Was the R18 stained virus/entry experiment done double blinded? This seems pretty subjective and would merit more valid experimentation. How many fields were enumerated? This manuscript would benefit to have a more “traditional” HCMV entry experiment (i.e., FL-labeled glycoprotein in the virion then measure attachment at 4C). It may be brighter than the R18 labeled virions. Confocal microscopy to visualize? Why not a traditional plaque reduction assay instead of relying on microscopy?

Thank you for the recommendation. To address this issue, we have done the plaque reduction assay with native PpFucCS at the same range of concentration but at a higher MOI of 3. The figures are added to the supplement.

4) Supplemental Figure: hard to tell whether the viruses are attached or not. The counter stain is the nucleus, and it is hard to visualize the outer membrane where the virus should be attaching. It is confusing why there is virus at 40 min and no virus at 60 min with treatment. This relates to Fig. 3. They need to have parallel time points with treated and untreated.

R18 is a lipophilic cation and HCMV virions labeled with high concentrations of R18 have fluorescence that is highly self-quenched and hence, we cannot see any fluorescence when the viruses are not attached. Only upon fusion of the HCMV particles with the HFF cell membranes, it exhibits a fusion-associated increase in the fluorescence. Hence, even though we have not overlaid the bright field images, we can still interpret that the virus has attached/ penetrated.

The figure 3B. is only the representation of a single field. When we took the mean of different fields as seen in the bar plots of figure 3A, the difference was not significant when compared by student t test in GraphPad Prism 9. Please see the figure below:

In case of mock treated samples, our aim was to prove that PpFucCS inhibits the cell entry (and not the kinetics) and hence just the 60 minutes reading would be sufficient because by 1 hour most of the virions get attached to the cell surface.

5) The differences in the fractions are interesting and should be explored further and may indicate how these novel glycans are preventing infection. Why does Fraction 1 and the depolymerized glycan still prevent attachment (Fig. 2) while 2-4 do not?

As stated in the discussion, the native high MW structure is necessary to achieve maximum HMCV inhibition. Fraction 1 is the heaviest fraction based on the molecular weight and hydrolyzed PpFucCS still retain fraction 1, thus some of the native structure may be maintained in these heavier fractions. In our findings, the decrease of chain length from Fr1 to Fr2 was significant enough to impact on the antiviral action, in which the oligosaccharide preparations with the longest chain lengths (HdPpFucCS and Fr1) are active as anti-HCMV.

6) They say in line 217 that “structural features play a crucial role in” anti CMV activity but they also say that “Fr1-3 differ in size, they have similar structures” (line 210-211). How do they reconcile this? Maybe they are referring to sulfations? If so, they need further experimentation to prove this.

The biomedical effects of marine sulfated glycans not only depend on one structural feature but both MW distribution as well as sulfation content, sulfation pattern and monosaccharide composition. Hence it is a combination of structural properties which determine the anti-viral effect of a marine sulfate glycan. A clear explanation and example about this were already in the text (see last paragraph of discussion). In this paragraph we discuss the different outcomes of BoSG and PpFucCS fragments as anticoagulant and anti-SARS-CoV-2 agents. In the case of BoSG, fragments of very low MW were capable to retain anti-SARS-CoV-2 activity as opposed to fragments of PpFucCS. BoSG is a sulfated galactan while PpFucCS is a fucosylated chondroitin sulfate. In our sentence, we meant that, although MW of PpFucCS is playing a primary role in the anti-HCMV activity, this role has been also mediated by other structural features of PpFuCS, such as fucosyl branching, GAG backbone, and sulfation pattern at the branching fucoses.

Round 2

Reviewer 2 Report

All issues I asked have been addressed.

Author Response

Thank you for the confirmation.